

# RNA-seq reveals more consistent reference genes for gene expression studies in human non-melanoma skin cancers

Van L.T. Hoang[1,*], Lisa N. Tom[1,*], Xiu-Cheng Quek[2,3,*], Jean-Marie Tan[1], Elizabeth J. Payne[1], Lynlee L. Lin[1], Sudipta Sinnya[1], Anthony P. Raphael[1,4], Duncan Lambie[5], Ian H. Frazer[6], Marcel E. Dinger[2,3], H. Peter Soyer[1] and Tarl W. Prow[1,7]

[1] Dermatology Research Centre, Diamantina Institute, Translational Research Institute, Princess Alexandra Hospital, The University of Queensland, Brisbane, Queensland, Australia
[2] Garvan Institute of Medical Research, Sydney, New South Wales, Australia
[3] St Vincent's Clinical School, University of New South Wales, Sydney, New South Wales, Australia
[4] Wellman Center for Photomedicine, Harvard Medical School, Massachusetts General Hospital, Boston, MA, USA
[5] Department of Anatomical Pathology, Princess Alexandra Hospital, Brisbane, Queensland, Australia
[6] Diamantina Institute, Translational Research Institute, Princess Alexandra Hospital, The University of Queensland, Brisbane, Queensland, Australia
[7] Biomaterials Engineering and Nanomedicine Strand, Future Industries Institute, University of South Australia, Mawson Lakes, Australia
* These authors contributed equally to this work.

Corresponding author
Tarl W. Prow,
Tarl.Prow@unisa.edu.au

## ABSTRACT

Identification of appropriate reference genes (RGs) is critical to accurate data interpretation in quantitative real-time PCR (qPCR) experiments. In this study, we have utilised next generation RNA sequencing (RNA-seq) to analyse the transcriptome of a panel of non-melanoma skin cancer lesions, identifying genes that are consistently expressed across all samples. Genes encoding ribosomal proteins were amongst the most stable in this dataset. Validation of this RNA-seq data was examined using qPCR to confirm the suitability of a set of highly stable genes for use as qPCR RGs. These genes will provide a valuable resource for the normalisation of qPCR data for the analysis of non-melanoma skin cancer.

## INTRODUCTION

There is a growing need for identification of biomarkers of non-melanoma skin cancer (NMSC) for accurate diagnoses of skin lesions, and to predict progression and patient response to novel treatments. Quantitative real-time PCR (qPCR) is an integral technique for gene expression analysis in dermatology research, due to its high sensitivity and specificity. The expression of target genes is calculated relative to reference genes (RGs) that are stably expressed. An ideal reference should be uniformly expressed in all samples within the given experiment. Historically, selection of RGs for qPCR studies has been

arbitrary, with researchers commonly selecting genes such as 18S rRNA, GAPDH, and Actin without experimental validation while making the assumption that they are stably expressed across tissues. However, we now know that in many instances these commonly used RGs exhibit tissue and treatment specific variability (*Chari et al., 2010*; *De Jonge et al., 2007*). A previous preliminary study on a number of cell lines and tumour versus matched normal tissue samples showed that inappropriate choice of RGs may lead to errors when interpreting experiments involving quantitation of gene expression (*Janssens et al., 2004*).

Validation of RGs tailored for individual experimental conditions is therefore a necessity before commencement of gene expression studies (*Bustin et al., 2009*). Use of a RG whose expression is variable or changes as a result of treatment conditions invariably leads to inaccurate and misleading results. It is therefore strongly recommended in the MIQE guidelines (minimum information for publication of qPCR experiments) that suitable RGs be determined for individual experimental conditions (*Bustin et al., 2009*). Selecting suitable RGs is not straight forward, and as a result researchers increasingly are turning to transcriptome profiling data to identify genes that are suitable for their tissue of interest.

Analysis of gene expression patterns in skin lesions by whole transcriptome sequencing (RNA-seq) is a powerful technique for the analysis of gene expression profiles (*Berger et al., 2010*; *Jabbari et al., 2012*; *Wagle et al., 2014*). RNA-seq allows accurate measurement of gene expression levels with a large dynamic range of expression and high signal-to-noise ratio. More importantly, RNA-seq, unlike probe-based assays (such as microarrays), provides an unbiased view of the transcriptome. As such, RNA-seq is an ideal strategy for identifying stably expressed genes suitable for use as qPCR RGs. The identification of stably expressed RGs in NMSC and precancerous lesions is essential to facilitate gene expression studies.

We have utilised next generation transcriptome profiling by RNA-seq to identify a list of candidate genes that exhibit very low variability across a range of NMSC lesions comprising intraepidermal carcinoma (IEC), squamous cell carcinoma (SCC) and precancerous lesions-actinic keratosis (AK). The stability of these candidate genes was validated by qPCR on a panel of NMSC lesions, including AK, IEC, SCC, basal cell carcinoma (BCC) and benign seborrheic keratosis (SK). Using geNorm and Normfinder analyses, we have determined a stable combination of genes as qPCR RGs specific for skin samples. We demonstrated the importance of accurate RG selection by performing relative quantitation analysis for several targeted gene expression profiles in non-photodamaged skin, AK and SCC lesions where normalisation was performed using either new RGs together or traditional RG GAPDH.

## MATERIALS AND METHODS

### Patient samples

Skin lesions and non-photodamaged skin tissue samples were collected from patients at the Dermatology Department at the Princess Alexandra Hospital. The study was approved by Metro South Human Research Ethics Committee and The University of Queensland Human Research Ethics Committee (HREC-11-QPAH-236, HREC-11-QPAH-477, HREC-12-QPAH-217, and HREC-12-QPAH-25). Written, informed consent was obtained from

all patients prior to participation. Following biopsy, tissues were immersed in RNA later (Life Technologies, Carlsbad, CA) and stored at −80 °C until required. All samples were sectioned and processed according to routine protocol in the Department of Anatomical Pathology located in Princess Alexandra Hospital.

## RNA isolation and cDNA synthesis

RNA isolation was performed using the Qiagen RNeasy Plus Mini kit (Qiagen GmbH, Hilden, Germany). Briefly, tissue samples were cut into small pieces, and transferred into 1.5 mL tube containing lysing matrix D (MP Biomedicals, Santa Ana, CA, USA) and 600 uL buffer RLT containing 1% beta-mercaptoethanol and homogenised using a Fast Prep benchtop homogeniser (MP Biomedicals, Santa Ana, CA, USA). Samples were spun five times using setting 6.5 for 30 s each, and chilled on ice between spins. Lysate was removed and transferred to a fresh tube. For unfixed BCC samples embedded in OCT, $20 \times 10 \, \mu m$ sections were cut and placed into 600 $\mu L$ buffer RLT, and homogenised using a 18.5 gauge blunt needle attached to an RNAse free syringe and resuspended five times. The remaining RNA extraction steps were performed as above. RNA concentration was measured using the Qubit fluorometer (Life Technologies, Carlsbad, CA, USA) and RNA integrity determined using the 2100 Bioanalyser (Agilent Technologies, Palo Alto, CA, USA) on RNA Pico chips. The minimum acceptable quality for RNA for analysis by qPCR was RIN >6. Complementary DNA (cDNA) was synthesised from 200 ng total RNA using the Sensifast cDNA kit (Bioline, London, UK) as per the manufacturer's instructions.

## Library preparation and RNA-seq

RNA-seq libraries of poly (A) RNA from 500 ng total RNA obtained from AK, IEC and SCC and SK samples, were generated using the TruSeq unstranded mRNA library prep KIT for Illumina multiplexed sequencing (Illumina, San Diego, CA, USA). TruSeq stranded mRNA library preparation kit was used to generate poly (A) RNA libraries for RNA obtained from normal skin samples (Illumina, San Diego, CA, USA). Libraries were sequenced (100 bp, paired-end) on the Illumina HiSeq 2500 platform and FASTQ files were analysed.

## Bioinformatics pipeline

Sequencing data were checked for sequencing quality by FASTQC (http://www.bioinformatics.babraham.ac.uk/projects/fastqc/). Adaptors and poor quality sequences were then removed using Trim Galore v0.3.7 (∼6% of reads removed) (http://www.bioinformatics.babraham.ac.uk/projects/trim_galore/). An average of 39.1 million trimmed reads pairs were then aligned using Tophat (V 2.1), using the unstranded protocol, against the Human Genome (hg19 build) guided with a human transcriptome generated from the GENCODE Gene annotation version 19 (Table S1) (*Harrow et al., 2012*; *Kim et al., 2013*). Quantification of gene expression based on counting the overlaps of mapped reads with genes annotated in the GENCODE gene annotation v19 using HTSeq (Version 0.6.1p2) (*Anders, Pyl & Huber, 2015*; *Harrow et al., 2012*) (Table S2). Read counts were then normalised using trimmed mean normalization method (TMM) using bioconductor's edgeR package (*Robinson, McCarthy & Smyth, 2010*; *Robinson & Oshlack, 2010*). Based on the Gencode gene model, an expression level of 57,278 genes was counted. Read counts

for the remaining 57,278 genes were then transformed into Transcripts Per Kilobase Million (TPM) values as previously described (*Wagner, Kin & Lynch, 2012*) (Gene length information found in Table S3).

To ensure that RGs can be detected in any of the samples, genes with any samples without detectable expression level (counts < 0) were discarded (38,670 genes discarded).

## Statistical analysis for identification of RGs from RNA-seq

The mean TPM value of a gene is calculated by taking the average of the gene expression across all samples. The CoV was measured by taking the standard deviation of expression value (TPM) for a given gene divided by its mean. The ratio of the maximum to minimum (MFC) was calculated using the largest TPM value divided by the smallest TPM value. A product score (MFC-CoV) was calculated based on the multiplication of CoV and MFC value for each gene. Biological Coefficient of Variance (BCV) was obtained by calculating the square root of the tag wise dispersion measured using edgeR bioconductor package (*Robinson, McCarthy & Smyth, 2010*) normalized counts per million. To facilitate exploration of the statistical analysis, data visualisation of the analysis can be found on http://skinref-dev.dingerlab.org/ and the raw data on ArrayExpress (https://www.ebi.ac.uk/arrayexpress/experiments/E-MTAB-5678/).

## qPCR

The primers for qPCR reactions were designed using NCBI Primer BLAST (http://www.ncbi.nlm.nih.gov/tools/primer-blast/) (refer to Table 1). Primers were designed to span intron boundaries to avoid amplification of genomic DNA and to amplify all isoforms known to each gene based on the NCBI Reference Sequence Database (RefSeq). Primers were synthesized by Sigma-Aldrich (Castle Hill, Australia). qPCR reactions were performed in triplicate using 1 μL diluted cDNA template in a 10 μL total volume. Reactions were performed in 384-well plate format on the ABI Viia7 Real-Time PCR system (Life Technologies, Carlsbad, CA, USA) using Sensifast SYBR Lo-rox mastermix (Bioline, London, UK). A 2-step cycling protocol was performed, comprising an initial 95-degree polymerase activation for 2 min, followed by 40 cycles of 95 degrees for 5 s, then 60 degrees for 20 s. The comparative Ct (ΔΔCt) method was used for data normalisation.

## Measurement of novel RG stability

The RG stability was calculated using the geNORM algorithm (*Andersen, Jensen & Ørntoft, 2004*), which is integrated with qbase+ software (Biogazelle, Gent, Belgium) and Normfinder software (*Vandesompele et al., 2002*) (available as an add-in for Microsoft Word at http://moma.dk/normfinder-software).

## Statistical analysis of qPCR data

Statistical tests used have been described in each figure legend. Data analysis was performed using GraphPad Prism version 5.04 for Windows (GraphPad Software, Inc., La Jolla, CA, USA).

**Table 1  Reference gene qPCR primers designed using NCBI Primer BLAST.**

| Gene | Accession number | Forward primer | Reverse primer | Amplicon size (bp) |
|---|---|---|---|---|
| RPL9 | NM_000661.4 | CTGCGTCTACTGCGAGAATGA | CACGATAACTGTGCGTCCCT | 98 |
| RPL38 | NM_000999.3 | GCCATGCCTCGGAAAATTG | CCAGGGTGTAAAGGTATCTGC | 139 |
| RPL11 | NM_000975.3 | AGAAGGGTCTAAAGGTGCGG | AGTCCAGGCCGTAGATACCA | 138 |
| RPL23 | NM_000978.3 | TCCAGCAGTGGTCATTCGAC | GCAGAACCTTTCATCTCGCC | 117 |
| EEF1B2 | NM_001959.3 | AGTATTTGAAGCCGTGTCCAG | ACATCGGCAGGACCATATTTG | 144 |
| RPS27A | NM_002954.5 | ACCACTCCCAAGAAGAATAAGC | ACTTGCCATAAACACCCCAG | 147 |
| RPL7A | NM_000972.2 | GGCATTGGACAGGACATCCA | AGGCACTTTCAGCCGCTTAT | 114 |
| RPS13 | NM_001017.2 | TCCCCACTTGGTTGAAGTTGA | AGGAGTAAGGCCCTTCTTGG | 77 |
| EEF1A1 | NM_001402.5 | GAAAGCTGAGCGTGAACGTG | AGTCAGCCTGAGATGTCCCT | 143 |
| RPLP0 | NM_001002.3 | ATCAACGGGTACAAACGAGTC | CAGATGGATCAGCCAAGAAGG | 97 |
| GAPDH | NM_002046.5 | CCCACTCCTCCACCTTTGAC | TTCCTCTTGTGCTCTTGCTG | 180 |
| HPRT1 | NM_000194.2 | TGCTGAGGATTTGGAAAGGG | ACAGAGGGCTACAATGTGATG | 115 |
| ACTB | NM_001101.3 | ACCTTCTACAATGAGCTGCG | CCTGGATAGCAACGTACATGG | 148 |

# RESULTS

## Identification of novel candidate RGs

To identify RG specific for NMSC and precancerous lesions, we first performed gene expression profiling using RNA-seq on 13 AK, seven IEC, five SCC lesions and four non-photo damaged skin samples. As previously described, a RG should show similar expression across samples, expressed at detectable levels and not display any exceptional expression in any of the samples (*De Jonge et al., 2007*; *Eisenberg & Levanon, 2013*). As with previous studies on identification of housekeeping genes, to identify genes that fall within these criteria, we measured the mean expression, CoV and the MFC for each gene within the dataset (*De Jonge et al., 2007*; *Eisenberg & Levanon, 2013*) (Table S4). The CoV measures variability within samples and the MFC is the ratio of the maximum and minimum TPM values for a given gene.

Ideally, a RG candidate should have a low CoV and MFC value and is expressed at detectable levels. As such, we generated a score based on the product of CoV and MFC value for each gene (MFC-CoV). We then obtained an initial list of 4643 gene candidates with a MFC-CoV that falls below the lower quartile value of MFC-CoV ≤1.63084, (Table S7). KEGG Pathway enrichment analysis on this initial gene list using geneSCF (*Subhash & Kanduri, 2016*) revealed that most of these stably expressed genes were involved in maintaining critical cellular activities including the citrate cycle, proteasome, RNA polymerase, protein export, RNA transport and Ribosome activities (B.H *FDR* Value < 1.8e−30) (*Huang da, Sherman & Lempicki, 2009*) (Fig. 1C and Tables S5–S6).

To identify RGs specific for NMSC and precancerous lesions, we shortlisted 10 candidate genes from the enriched pathways involved in critical cellular activity for further validation with qPCR. In addition, we selected candidates with functions well described in literature. For instance, we chose the RPLP0 gene, whose function is not only well known for different cell and tissue types but also shown to be a suitable RG for research in the differentiation

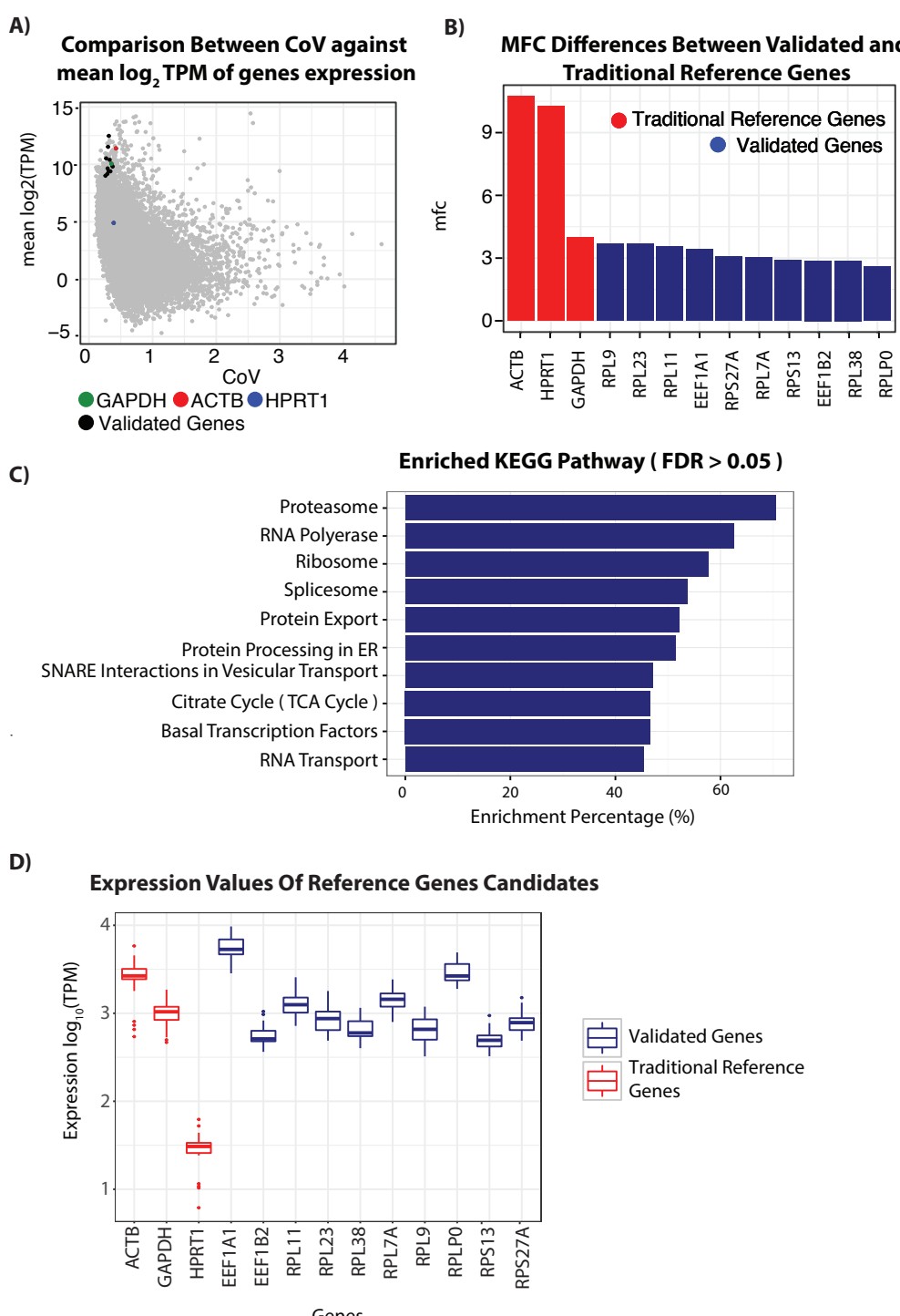

**Figure 1 RNA-seq analysis of genes and candidate reference genes.** (A) Scatterplot comparing Coefficient of variation (CoV) values against mean expression values (log2) transformed and represented in Transcript per kilobases million (TPM) for genes detected using the RNA-seq of patient derived skin samples. Each gene is represented by a single dot. Genes selected for validation to 

**Figure 1 (...continued)**
function as reference genes in non-melanoma skin cancers (NMSC) and precancerous lesions are shown in Black. Reference genes commonly used in the literature, ACTB (Red), GAPDH (Blue) and HPRT1 (Green), are also highlighted for comparisons. (B) Comparison of maximum fold change score in gene expression of candidate reference genes and traditional reference genes. (C) Results of KEGG pathway enrichment analysis conducted with a list of 3,714 genes found with product score between each gene's MFC and CoV score below the lower quantile. Enrichment percentage is defined as the percentage of genes in the pathway that are overlaps with genes in our list. (D) Boxplot showing expression value from RNASeq experiment of 29 skin lesions of selected reference genes candidate (blue) with commonly used housekeeping genes ACTB, GAPHD and HPRT1 (red).

**Table 2   RNA-seq scoring of selected candidate reference genes and commonly used reference genes.**
RNA-seq scoring of selected candidate reference genes and commonly used reference genes, ranked on CoV (coefficient of variation) score, mean, mean expression value, MFC, maximum fold change calculated using transcript per million values. Candidates are ranked from the smallest to largest CoV values.

| Gene Symbol | CoV | Mean | MFC |
|---|---|---|---|
| RPS13 | 0.25929018 | 471.6328 | 2.9453 |
| RPL7A | 0.270464492 | 1052.6613 | 3.0589 |
| EEF1B2 | 0.289246663 | 569.7829 | 2.9703 |
| RPS27A | 0.292810337 | 625.2571 | 3.2915 |
| RPLP0 | 0.300470977 | 166.6758 | 3.0214 |
| RPL38 | 0.307936407 | 1336.0407 | 2.7401 |
| EEF1A1 | 0.312400695 | 2809.6306 | 3.5469 |
| RPL11 | 0.325188306 | 926.3678 | 3.8090 |
| RPL9 | 0.336714133 | 489.1896 | 3.9220 |
| GAPDH | 0.352954726 | 865.5278 | 4.2344 |
| RPL23 | 0.372401947 | 241.0372 | 3.9124 |
| HPRT1 | 0.3960 | 35.4396 | 11.0423 |
| ACTB | 0.4260 | 2005.6012 | 10.9588 |

of human epidermal keratinocytes (*Bar, Bar & Lehmann, 2009*). Finally, to distinguish mRNA from genomic DNA, we selected multi-exonic genes as candidates to aid design of primers across intron boundaries.

To determine the performance of our gene candidates, we compared the CoV, MFC and mean expression value of our 10 RG candidates with three commonly used RGs in qPCR analysis of skin—ACTB, GAPDH and HPRT1 (Table 2) (*De Kok et al., 2005*). With the exception of RPL23, our candidate RGs had a lower CoV-MFC compared to the traditional RGs ACTB, HPRT1 and GAPDH (Table 2 and Figs. 1A and 1B). RPL23 had a lower MFC but a higher CoV than GAPDH. We also calculated the BCV and found that our candidate genes fall below the mean (BCV is similar to that of the traditional house keeping genes (Fig. S1B and Table S2).

Data visualisation of TPM, CoV and MFC metrics of the calculated genes in this study have been made available online (http://skinref-dev.dingerlab.org/) as a resource to facilitate other investigators in the community to explore the datasets and identify their candidate RG of interest (Fig. S2).

## qPCR validation of new RGs

To validate and extend our findings from the RNA-seq analysis, we conducted qPCR on our 10 candidate RGs in addition to the three commonly used skin RGs ACTB, GAPDH, and HPRT1 on samples derived from a diversity of skin conditions within the spectrum of NMSC (SCC and precursors and BCC) (refer to Table 1 for list of gene primers). A total of 24 samples were tested and they include AK (4), SCC (3), SK (3), BCC (4), IEC (5), and non-photodamaged skin (5). The use of independent samples across a variety of NMSC skin lesions also minimises any patient specific expression biases of our RGs. This was important as some of the lesions used in RNA-seq was derived from the same patient and readings of MFC, CoV and BCV among our samples could exhibit patient-specific biases. Results from the qPCR were analysed using geNorm (*Andersen, Jensen & Ørntoft, 2004*) within Qbase+ software (Biogazelle) and Normfinder to determine the consistency of expression values among the samples for each candidate RG (*Vandesompele et al., 2002*).

Statistically, geNorm conduct pairwise variation (V) analysis to identify genes with the least variance between samples and is denoted as 'stability' ($M$) values. In general, lower $M$ values indicate lower variance in expression value among samples and genes with $M$ values $\leq 0.5$ are associated with homogeneous samples. Remarkably, all of our 10 RG candidates had $M$ values $\leq 0.5$ with RPLP0, RPL7A, RPL23, RPS27A and RPL38 ranked in the top five genes for geNorm $M$ value/Stability value. In addition, to eliminate errors related to the usage of a single housekeeping gene, it is common practice to use two or more housekeeping genes. By calculating the normalization factor based on the geometric mean of multiple control genes, we identified that we need only two of our RG candidates for accurate normalization (geNorm $V$, $V2/3 = 0.084$). $V$ values of $<0.15$ indicate acceptable stability of the RG combination, indicating no further need for additional RGs. Amongst our RG candidates, the pair of genes with optimal normalization factor was RPL38 and RPS27A, which demonstrated the lowest $M$ values (0.257 and 0.265 respectively) (Fig. 2A).

In addition, Normfinder analysis was performed for the same dataset. Normfinder analysis performs estimation of both intra- and intergroup expression variation for each subgroup of samples (lesion types), with output given as a Stability Value. The most stable candidate was RPL7A, and the best combination of genes was RPL7A and RPLP0 (Fig. 2B). Overall trends between geNorm and Normfinder analyses were similar. In both formats, the traditional RGs ACTB, GAPDH and HPRT1 were ranked as having the most variability in gene expression across the groups (increased stability values), and the genes RPLP0, RPL7A, RPL23, RPS27A and RPL38 ranked in the top five genes for geNorm $M$ value/Stability value.

To demonstrate the significance of our findings in NMSC research, we investigated the difference in expression of keratin 17 (KRT17) in AK between normalization using our candidate RGs and normalization with GAPHD (Fig. 3). When using GAPDH as calibrator, there was an approximate 2-fold increase in levels of KTR17 when comparing non-photodamaged skin to AK (Fig. 3A) or approximate 3-fold increase in SCC (Fig. 3B). However, this fold change was significantly higher at approximately 7-fold for AK and approximately 12-fold for SCC when using the combination of RPL32 and RPS27A or either one of them as RG ($P < 0.05$). There was no statistical difference between data normalised

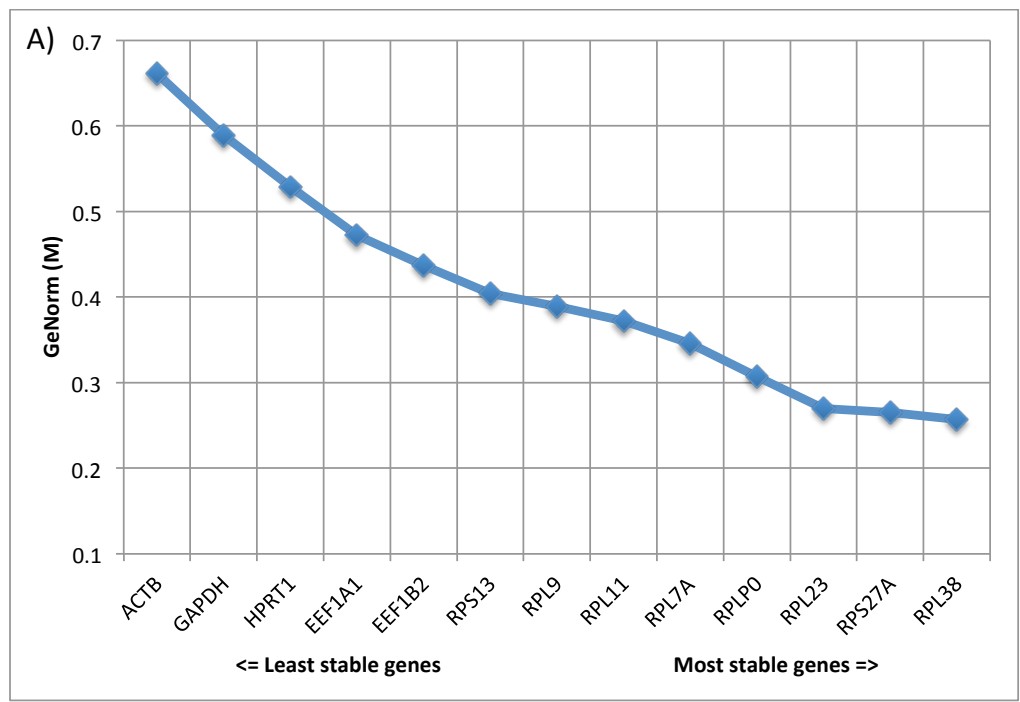

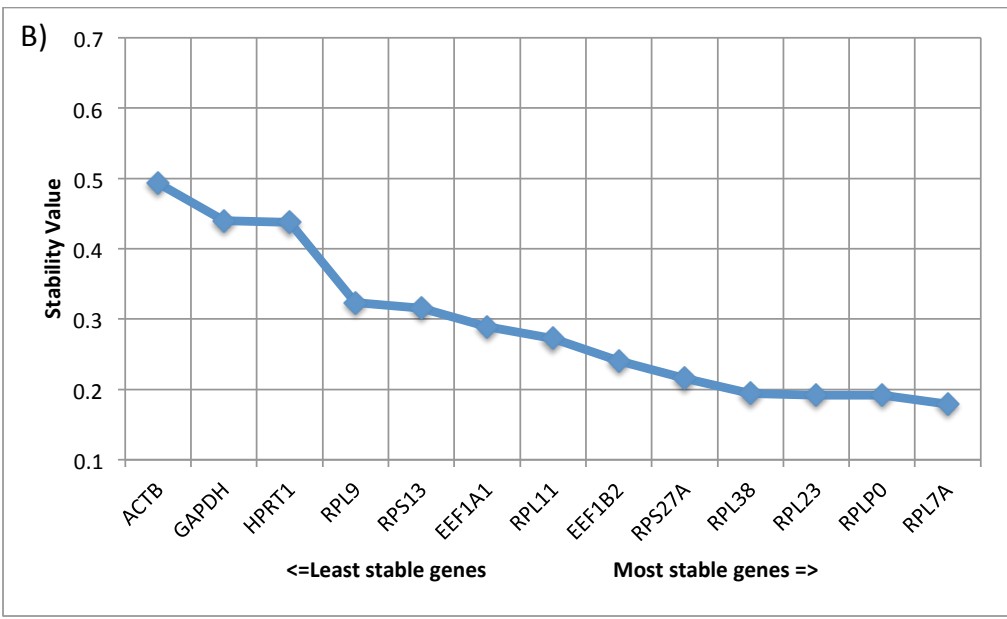

**Figure 2** **Comparison of expression stability using GeNorm and Normfinder.** (A) Average expression stability of reference targets (geNorm). geNorm *M* value, an indicator of gene expression stability, was determined using the geNorm algorithm. Decreasing values correlate with smaller variations in gene expression levels across lesion groups AK, SCC, SK, BCC, IEC, and healthy skin. (B) Average expression stability of reference targets (Normfinder). Stability values were determined for each gene using the Normfinder algorithm. Decreasing values correlate with smaller variations in gene expression levels across lesion groups AK, SCC, SK, BCC, IEC, and healthy skin.

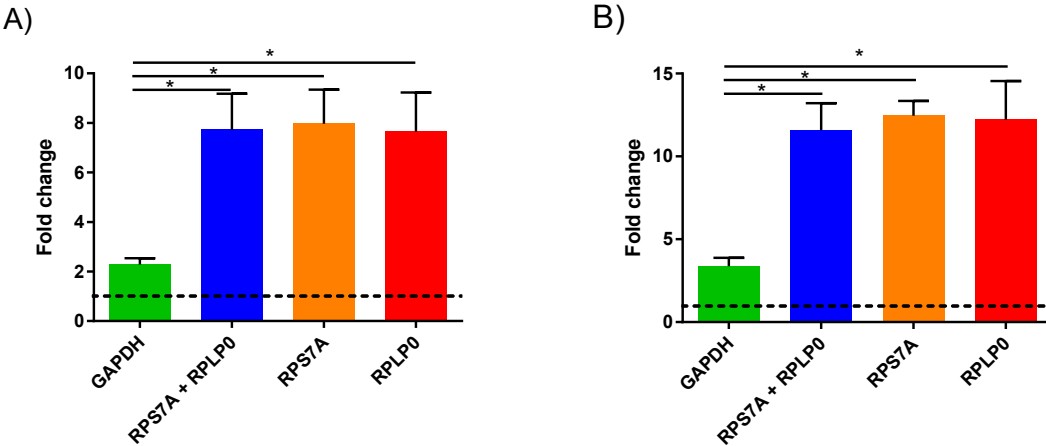

**Figure 3** **KRT17 levels in precancerous and lesional NMSC.** Comparison of relative quantitation analysis of KRT17 levels in AK (A) and SCC (B) lesions using either RPS7A/RPLP0 or GAPDH as the reference gene relative to non-photodamaged skin. Data are presented as mean ± SEM, $n = 3$, ∗ indicates $P < 0.05$; one-way ANOVA and Turkey post-test.

with RPLP0, RPS7A or both of the two candidate genes. This result demonstrates that use of a RG, which is not stably expressed, can lead to inaccurate data, particularly in instances where the relative fold change is subtle.

## DISCUSSION

The selection of appropriate RGs is of critical importance for accurate quantification of gene expression levels using qPCR. Our results concur with previous studies reporting that RNA-seq is an effective method for the identification of stably expressed transcripts for applications in qPCR. Through qPCR validation we demonstrate that transcriptome analysis by RNA-seq is a reliable strategy for identification of genes with low variability. To the best of our knowledge, this is the first study to identify suitable RGs for use in studies of pre-cancerous lesions and NMSC. Our data demonstrate that the RG candidates selected for validation are stably expressed in these lesions, showing strong stability in gene expression between different types of skin cancer lesions and non-photo-damaged skin. Results suggest that our RNA-seq dataset is a valuable resource to assemble a shortlist of candidates for validation by qPCR prior to commencement of gene expression studies in NMSC and sun-damaged skin. Our RNA-seq data identified many RPL and RPS genes, which encode structural proteins associated with ribosome biosynthesis, as highly stable. This finding is in agreement with previous studies demonstrating RPL genes as some of the least variable across a wide range of cell and tissue types. In a meta-analysis of over 13,000 human gene arrays, 13 of the top 15 genes identified were ribosomal structural proteins (*De Jonge et al., 2007*). The need for stability in this group of genes is logical given that ribosome biogenesis is a tightly regulated process that is critical for fundamental cellular functions including cell growth and division.

We evaluated the stability of ten RGs in various NMSC samples. Results showed small differences in the recommended RG combinations between geNorm and Normfinder

analysis outputs, but the overall stability of our candidate RGs was shown to be consistent in both analyses (Fig. 2). This effect is likely due to the way these algorithms are designed, each utilising a different method to determine the most stable gene combinations. In the case of geNorm, the algorithm uses pairwise correlation to determine stability, using the assumption that genes showing similar expression patterns are likely to also reflect mRNA (cDNA) levels. BestKeeper is another commonly used normalisation algorithm that is based on pairwise correlation (*Pfaffl et al., 2004*). A limitation of this type of normalisation process is that genes, which demonstrate co-ordinate regulation, are likely to be ranked highly, even if they are not truly stable. Normfinder is an alternative algorithm, which uses a mathematical model-based approach, which allows estimation of both intra- and intergroup expression variation to calculate a stability value. Due to this variability, it is a wise strategy to use more than one algorithm to confirm the most appropriate RGs. In our case, there is a very small variation between the highest-ranking candidates for both analytical methods. In general, any of these top ranked genes RPL38, RPL23, RPS27A, RPL7A and RPLP0 are suitable RGs for use in NMSC and precancerous lesions. By contrast, GAPDH or ACTB, which are widely used as RGs are not suitable in this type of cancer as their expression is significantly different in non-photodamaged skin and the different type of NMSC. This finding is similar to the results of a recent study that recommended not using GAPDH for normalization purposes when analysing RNA expression in human keratinocytes (*Beer et al., 2015*).

To observe the impact of RG stability on relative quantitation analysis, we analysed the levels of keratin KTR17 in non-photodamaged skin, SCC, and AK lesions using either GAPDH or our most stable combination as determined by Normfinder analysis, RPS7A and RPLP0. KRT17 together with KRT16 and KRT6 are involved in keratinocyte differentiation and skin cancer (*Hameetman et al., 2013*). It was previously reported that intermediate filament keratins in SCC lesions compared to non-photodamaged skin were upregulated (*Hameetman et al., 2013*; *Hudson et al., 2010*). In this study, we found that using different calibrators significantly altered the comparison result. The upregulation of KRT17 in AK and SCC lesions was even more pronounced with our candidate RGs. These results demonstrate that the latter set of genes can give better power in distinguishing KRT17 expression between healthy and AK or healthy and SCC.

However, it should be noted that despite the high stability of our candidate RGs across a range of different skin lesions, these lesions were not exposed to any treatments such as topically applied medications, which could potentially affect their expression. A literature search should be performed prior to the commencement of the study to eliminate RGs that will potentially be affected by treatment conditions. As it is unlikely that any gene is stable across all possible experimental conditions, validation should be performed for each treatment, and in general, two or more RGs should be used to reduce the impact of any variability. For our subset of validated RGs, many are genes encoding ribosomal structural proteins. Caution should be used if considering these RGs where treatment conditions have been demonstrated to result in nucleolar stress (*Nosrati, Kapoor & Kumar, 2015*). In this instance, selection and validation of genes with a different functional classification, such as EEF1A1 or EEF1B2, or derived from our initial long list of stable genes would be a

logical strategy. Furthermore, while our RG candidates from RNA-seq has been validated to outperform traditional RGs, there were several limitations in our RNA-seq analysis. Due to the mix of stranded and unstranded RNA-seq samples used, to allow fair comparison across the samples, strand specific expression of these genes was not taken into account. Expression quantification of these RGs was conducted on a gene level, which may differ at the isoform level. Regardless, using qPCR we independently validated our RG candidates.

## CONCLUSIONS

In this study, we utilized whole transcriptome RNA-seq to analyze healthy skin, precancerous and lesional NMSC for the purpose of identifying RGs, which are consistently expressed across all samples. To identify genes that fall within these criteria, we measured the mean expression, CoV and the MFC for each gene within the dataset. This resulted in the identification of 100 highly stable genes. To further refine the genes specific for precancerous and NMSC lesions, we then shortlisted 10 candidate genes for further validation with qPCR. These 10 candidate genes were selected based on cut-off values set lower or higher than both the mean and median values of the transcriptome. We determined that the genes RPL38, RPL23, RPS27A, RPL7A and RPLP0, which encode structural proteins associated with ribosome biosynthesis are the most suitable RGs for use in NMSC and precancerous lesions.

## ACKNOWLEDGEMENTS

We would like to acknowledge the cooperation and coordination between all the members involved within this multi-center study including the Dermatology Research Center, Garvan Institute of Medical Research, Department of Anatomical Pathology at the Princess Alexandra Hospital, the Diamantina Institute and assistance from those in the Institutes' sequencing facilities.

### Funding
This project was supported by Epiderm (Boronia Park, NSW 2111, Australia) and NHMRC Fellowships APP1109749 and APP1088318. The funders had no role in study design, data collection and analysis, decision to publish, or preparation of the manuscript.

### Grant Disclosures
The following grant information was disclosed by the authors:
Epiderm.
NHMRC Fellowships: APP1109749, APP1088318.

### Competing Interests
H. Peter Soyer and Tarl W. Prow are Academic Editors for PeerJ.

## Author Contributions

- Van L.T. Hoang and Lisa N. Tom conceived and designed the experiments, performed the experiments, analyzed the data, wrote the paper, prepared figures and/or tables, reviewed drafts of the paper.
- Xiu-Cheng Quek conceived and designed the experiments, performed the experiments, analyzed the data, contributed reagents/materials/analysis tools, wrote the paper, prepared figures and/or tables, reviewed drafts of the paper.
- Jean-Marie Tan, Elizabeth J. Payne, Lynlee L. Lin, Sudipta Sinnya and Duncan Lambie performed the experiments, contributed reagents/materials/analysis tools, reviewed drafts of the paper.
- Anthony P. Raphael wrote the paper, prepared figures and/or tables, reviewed drafts of the paper.
- Ian H. Frazer, Marcel E. Dinger and H. Peter Soyer conceived and designed the experiments, contributed reagents/materials/analysis tools, reviewed drafts of the paper.
- Tarl W. Prow conceived and designed the experiments, wrote the paper, prepared figures and/or tables, reviewed drafts of the paper.

## Human Ethics

The following information was supplied relating to ethical approvals (i.e., approving body and any reference numbers):

The study was approved by Metro South Human Research Ethics Committee and The University of Queensland Human Research Ethics Committee (HREC-11-QPAH-236, HREC-11-QPAH-477, HREC-12-QPAH-217, and HREC-12-QPAH-25).

## Data Availability

The raw data are available in the ArrayExpress database

Accession number E-MTAB-5678.

## Supplemental Information

Supplemental information for this article can be found online at http://dx.doi.org/10.7717/peerj.3631#supplemental-information.

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
