# Peer review of "RNA-seq reveals more consistent reference genes for gene expression studies in human non-melanoma skin cancers"

_PeerJ, doi:10.7717/peerj.3631_

## Round 0.1 · original submission · Major Revisions

The manuscript has been carefully evaluated by two external reviewers. We found merits in this interesting manuscript addressing a crucial point in RNASeq research. However, the reviewers raised some specific questions that would need to be addressed properly and that would consolidate your manuscript for publication. I would be glad to consider a revised version of the manuscript where all the concerns raised by the reviewers will be addressed point by point.

·

Basic reporting

The manuscript by Hoang et al seeks to identify stable reference genes in healthy skin and non-melanoma skin cancer lesions. The use of appropriate and stable reference genes is critical for design of low throughput analysis methods including real time PCR. The authors use next generation sequencing of identify genes which are consistently expressed across all samples. The stability of identified gene expression is further analyzed using qPCR. Two well known algorithms, GeNorm and Normfinder are used to example stability of identified genes, and to identify most stable combinations of RGs. The authors demonstrate the utility of this approach by using two of the identified RGs (RPL32 and RPS27A) to normalize levels of KTR17 expression between healthy skin and AK or SCC. Importantly they identify that normalizing to the identified RGs leads to a more dramatic fold change than when normalizing with GAPDH. Altogether, the manuscript is well written, clearly outlines the experimental approach and conclusions, and is accompanied by appropriate and clear figures.

Experimental design

The use of appropriate reference genes (RGs) is critical for the proper interpretation of low throughput assays such as qPCR. As an unbiased approach to identify RGs in healthy and non-melanoma skin lesions the authors perform RNASeq on 4 healthy skin samples, 12 AK, 7 IEC and SCC lesions. Line 156 should be modified to more clearly indicate the number of SCC lesions which were analyzed. The authors use appropriate bioinformatics approaches to address the issue of RG identification. Methods and conclusions are well described.

Validity of the findings

Overall, the data, as presented appears robust and appropriate analysis methods appear to have been used. The conclusions described in the manuscript are appropriately supported by the presented data. The authors are careful to point out that these RGs are applicable to skin lesions in the absence of drug treatment. Line 71 calls out the identified RGs as specific to ‘skin samples’ – it may be interesting to evaluate if the identified RGs can also be applied to melanoma samples; similarly, analysis of the expression of the identified RGs across different tissues (using publically available data) could be interesting, but is likely beyond the scope of the current manuscript.

It is recommended that the reader is directed to table 2 from the experimental methods section describing the qPCR. This would allow readers to more quickly find the sequence of primers to use these in their own work. Additionally, it may be beneficial to plot the raw values (colored by the type of sample) for each of the identified RGs and the historical RGs. This would serve as a compliment to the presented summary data and would assist the reader in evaluating the stability of the identified RGs across different lesion types. This reviewer was not able to find reference to deposition of the raw sequencing data in a public repository. This would be beneficial for other researchers hoping to utilize these data for similar studies.

Reviewer 2 ·

Basic reporting

In general, the paper was easy to read and clearly worded, although at times insufficient details lead to ambiguity. In the introduction, more information on how reference genes (RGs) are used for the normalisation of qPCR data would be useful to readers who are unfamiliar with the application. This would help readers understand the importance of finding a good set of RGs, and consequentially, the importance of this paper. (How can diagnosis results be affected by choice of poor RGs, or if no RGs are used?)

There are some inconsistencies in the use of abbreviations and capitalisations throughout the article. Sometimes previously abbreviated words are seen in the non-abbreviated form, e.g., “maximum fold change” and “coefficient of variation” is used in the unabbreviated forms in the conclusion when the abbreviations of MFC and CoV are used throughtout the article; or abbreviations are redefined repeatedly, e.g. “coefficient of variation (CoV)” in line 125 , “Coefficient of variation (CoV)” in Figure 1 caption, and “CoV (coefficient of variation)” in Table 1 caption. Also, in line 202, it should be “common practice” rather than “common practise”.

The caption for Table 2 can be expanded, and Figure 3 caption can be described in more detail (especially with regard to what comparison the “Fold change” represents, as well as “one-way ANOVA and Turkey post-test”).

Raw data is not supplied.

Experimental design

The overall experimental design appears to be promising but there is a lack in detail of how the experiment was carried out and analysed. For this reason, it is difficult to assess whether novel RGs were discovered using rigorous methods of investigation or not.

- Were all skin lesions and healthy skin tissue samples taken from different individuals? Did any of the patients contribute more than one sample that is used in downstream analysis? If so, was this factored into the analysis?
- Is there a reason why different library preparations were carried out between the skin lesions and healthy skin samples?
- In line 112, does “(~40 million reads)” refer to the number of reads per library? Please specify to avoid ambiguity.
- Were read counts normalised using the trimmed mean of M-values method within the edgeR software, or another normalisation method? Please specify (and cite “A scaling normalization method for differential expression analysis of RNA-seq data” by Robinson and Oshlack if it is the TMM method). Also, “edgeR” is the correct capitalization of the method, rather than “EdgeR”.
- In line 126-127, it would be more accurate to say that you have calculated the MFC value using the “ratio of the maximum to minimum value” rather than “minimum and maximum value”. Although it is not stated in the paper whether the MFC is defined as the max:min- or min:max-ratio, I assume that it is max:min since the results select for genes that have MFC<5. A clarification of this is necessary since all genes would have a MFC value of <5 if MFC=min:max, rather than MFC=max:min. Also, the two possible definitions of MFC would lead to a difference in whether it is the genes with low or high MFC value that are of interest.
- In line 156, how many SCC samples are used?
- How does GeNorm work out that only two of the RGs are required for accurate normalisation? This part of the experiment appears to be only carried out for the 13 genes that have been shown to be homogenous in expression. It sounds counter-intuitive for the GeNorm software to work out how to normalise the expression of biomarkers that are variable in expression by using genes (RGs) with homogenous expression only.

Validity of the findings

Since RNA-seq data is used here, it seems natural to use approaches that have been carefully considered and tailored to this data-type. The authors use CoV and MFC values to study the level of variation within genes. Why not examine the biological coefficient of variation (BCV) for genes instead, since the method for estimating these values have been specifically designed for count data in the edgeR software package that the authors are already using. Moreover, expression data (including RNA-seq data) is commonly analysed on the log-scale so that values are within a more manageable range. Raw RPKM values range from zero to tens-of-thousands; log2-RPKM values have a range of -15 to 15, or thereabouts. It seems sensible to use log2-RPKM values rather than RPKM values for analysis.

In Results, Identification of novel candidate RGs, it would be helpful to see a variation of plots showing what log10RPKM values, CoV and MFC values are for all the genes, with the 10 candidate genes highlighted (as well as ACTB, GAPDH and HRT1). This will give readers an idea of what the distribution and spread of these values are; whether the cut-off values chosen by the authors are sensible; and where the candidate genes sit relative to other genes that have not been selected as RGs for these summary statistics. In addition, one of the criteria for candidate gene selection is that “log10RPKM>1” (line 175). How are these values summarised for each gene? Is this the genewise mean log10RPKM value (or log10 of the genewise mean RPKM value)?

In Results, qPCR validation of new RGs, is the qPCR experiment carried out on samples taken from a different set of patients to those that are used in the RNA-seq experiement? More weight will be given to qPCR results from a different set of patients since the two sets of data would be independent of each other. If a large proportion of samples/patients overlap with the RNA-seq experiment, the qPCR results would show that the expression results can be translated across different experimental platforms, but it would not be too surprising to see that genes with low variability in the RNA-seq experiment are also “stable” in expression when measured by qPCR.

The qPCR validation appears to be carried out for 13 genes only (10 new RGs + three commonly used RGs). If this is the case, it does not seem surprising at all that 5 out of the 10 novel RGs are ranked at the top (lines 199-201). I pressume that some of these RGs also ranked at the bottom. To show that the 10 novel RGs rank well, it would be more useful to include a (large enough) set of randomly selected genes and show where the 10 novel RGs rank; and what the range of M values for the genes not selected as RGs relative to the M values for the RGs.

The authors show that the expression of KRT17 in healthy to AK (and healthy to SCC) are different after normalisation by GAPHD versus normalisation by the new RGs. Seeing that this study is about finding new RGs for use of normalisation, it would be interesting to see more examples of how gene expression differs based on the old and new RGs. This would show whether the novel RGs would really make a difference across large range of biomarkers, or if such striking differences are only observed in KRT17. From this one example, I would disagree that the “results demonstrate that use of a RG that is not stably expressed can lead to inaccurate data” (lines 227-228) or “wrong conclusions being drawn from qPCR results” (lines 279-280) seeing that both methods show the same result -- that KRT17 is upregulated. Unless it has been reported in previous studies, we do not know whether the true difference between the two groups is closer to the values of 2- and 3-fold increase (using GAPDH as a reference), or 7- and 12-fold increase (using the combination of RPL32 and RPS27A). It would be more accurate if instead the authors say that the latter set of genes can give better power in distinguishing KRT17 expression between healthy and AK (or healthy and SCC).

The authors have noted that the use of some RGs such as GAPDH have been selected “without experimental validation” (line 42) and “are not suitable in this type of cancer as their expression is significantly different in healthy skin and different type of NMSC” (lines 266-267). This makes the comparison of their 10 novel RGs against such genes not very convincing, since their genes simply “work better” relative to a set of genes that do not work well. Instead, as suggested earlier, the 10 novel genes should be compared against a larger set genes that are not selected as RGs to show that the 10 RGs have attributes that set them apart from the rest and are suited for use as RGs.

Additional comments

By using RNA-seq expression data, the authors present a novel set of RGs that can be used to normalise qPCR data for the analysis of non-melanoma skin cancer (NMSC). The methods outlined in the paper can be easily applied to studies other than those in NMSC, leading to the potential discovery of RGs in other areas. The idea behind the paper appears to be promising, however, several key sections are lacking detail and reasoning to properly comment on the validity of the paper’s findings and whether the study is valuable in terms of contribution to the community. The newly discovered RGs would be more convincing if the authors show evidence of how those genes stand out as RGs by plotting its distinct features (high expression, low variation, etc.) relative to the rest of the genes that are not selected as RGs. Moreover, I believe using methods that are tailored to RNA-seq data analysis could give more reliable results than the summary statistics that are used in this paper and that are not commonly used in RNA-seq data analysis.

---

## Round 0.2 · Minor Revisions

The manuscript has been substantially improved upon revision. There are still some minor issues to solve in terms of analyses and editing of the manuscript, as pointed out by the reviewer. We would be delighted to receive a revised version of the manuscript where the remaining issues will be addressed.

Reviewer 2 ·

Basic reporting

As noted in my previous review, inconsistencies and incorrect capitalisation can be found throughout the article. I appreciate the corrections that have been made to “CoV” and “MFC” since they were mentioned specifically, however other inconsistencies can still be found.
- The same gene is called “HRT1” (line 203 and Figure 1 caption), and at other times it is call “HPRT1” (line 204 and within Figure 1).
- An algorithm is called “GeNorm” (line 74), then called “geNORM” (line 160).
- GeNorm/geNORM has citation of “Andersen et al. 2014” in line 161, and then has a different citation, “Vandesompele et al. 2002” in line 221.
- RNA sequencing has been defined as “RNA-seq” in the paper (in Abstract), but line 173 uses “RNA-Seq” and Figure 1 caption uses both “RNA seq” and “RNASeq”.
- Line 129 “reference genes” can be abbreviated to “RGs” as defined earlier in the paper.
- Line 197 has citation “(16)” that is not linked to any references, and is also inconsistent in terms of citation styling.

A quick Google-search shows that the correct capitalisation of algorithm names and citations are:
- “geNorm” with citation as “Vandesompele et al. 2002”. The paper calls this “GeNorm” throughout most of the paper, and cites “Andersen et al. 2014” in line 161.
- “NormFinder” with citation as “Andersen et al. 2002”. The paper call this “Normfinder” throughout, and cites “Vandesompele et al. 2002” in line 163.
- As mentioned in the last review, “EdgeR” (lines 127 and 142) is officially capitalised as “edgeR”.

Other:
- Is “dermatology research itation” in line 40 a typo?
- “to a stably RG” in line 41 should perhaps be corrected to “to RGs that are stably expressed”.
- Line 138, “for a given gene by its mean” should perhaps be corrected to “for a given gene divided by its mean”.
- The plots displayed in the link on line 144 have permanent axis labels (Log10TPM on the x-axis, and CoV on the y-axis) that do not reflect the changes in values when selecting log10MFC, etc.
- Line 208 has a missing close bracket.

Although almost all papers will suffer from some minor grammatical errors and/or typos, but the extent to similar errors in this paper is quite pronounced. The authors should take more care in finding and correcting errors in text inconsistencies and referencing.

Experimental design

Comments to author’s rebuttal:

7) Did any of the patients contribute more than one sample that is used in downstream analysis? If so, was this factored into the analysis?
> There are patients who contributed to more than one samples, … this information are not factored into the analysis.

Whilst the exclusion of patient information in the analysis should make little impact on the conclusions of this paper, seeing that the new RGs have been validated by qPCR. However, authors should note that reported values for MFC, CoV and BCV could be somewhat deflated since sample measurements are not independent of each other.


8) Is there a reason why different library preparations were carried out between the skin lesions and healthy skin samples?
> The sequencing was done at two different occasions using library preparation protocols from the same company. Reads from both libraries were aligned using the unstranded protocol and sequenced within the same run.

The samples that were sequenced using a stranded protocol should most definitely use an alignment protocol that adjusts for its strandedness, rather than treating it as unstranded. A stranded protocol eliminates the possibility of sequencing genes on one of the strands so it makes little sense to assume that subsequent reads could come from genes on either of the strands, when you already know which strand it belongs to. Having said that, this should make little impact on the conclusions of this paper seeing that the RGs have been validated.

Validity of the findings

The paper discovers 10 novel RGs that are useful for the normalisation of qPCR data in NMSC studies. The 10 genes have been validated by qPCR and show attributes that are better than existing RGs. The novel set of genes look promising as RGs, and the addition of the plots found on http://skinref-dev.dingerlab.org provides a good illustration of where the RGs sit relative to other genes in the RNA-seq data analysis.

Other queries/suggestions:
i. Genes “without detectable expression level” have been removed (line 130). The criterion of “counts<0” has been specified (line 130). Have raw counts been transformed to a different scale? If so, they were not mentioned. TPM-transformation seems to occur after this step. However, if these are raw counts, it is not possible for counts to be less than zero. Are the genes removed if they have counts=0 rather than counts<0?
ii. TPM values have been added to the manuscript (lines 131-133). Authors should note that TPM values calculated after discarding genes may not accurately reflect the original sequencing depth of each sample, especially when almost 70% of genes are removed. Calculating TPM values prior to removal of genes would ensure accurate adjustment by sequencing depth.

Additional comments

Whilst the results and conclusions appear to be valid, some minor changes to other parts of the analyses could have provided evidence that is even more convincing in support of the paper’s conclusions. For example, more rigorous approaches (alignment using a stranded protocol and calculation of TPM values as suggested above), or more in depth analyses (factoring patient-sample contributions as suggested above).

---

## Round 0.3 · accepted · Accept

I am delighted to endorse this nice manuscript for publication on PeerJ.